# Learning Temporally-Consistent Representations for Data-Efficient Reinforcement Learning

## Abstract

Deep reinforcement learning (RL) agents that exist in high-dimensional state spaces, such as those composed of images, have interconnected learning burdens. Agents must learn an action-selection policy that completes their given task, which requires them to learn a representation of the state space that discerns between useful and useless information. The reward function is the only supervised feedback that RL agents receive, which causes a representation learning bottleneck that can manifest in poor sample efficiency. We present $k$-Step Latent (KSL), a new representation learning method that enforces temporal consistency of representations via a self-supervised auxiliary task wherein agents learn to recurrently predict action-conditioned representations of the state space. The state encoder learned by KSL produces low-dimensional representations that make optimization of the RL task more sample efficient. Altogether, KSL produces state-of-the-art results in both data efficiency and asymptotic performance in the popular PlaNet benchmark suite. Our analyses show that KSL produces encoders that generalize better to new tasks unseen during training, and its representations are more strongly tied to reward, are more invariant to perturbations in the state space, and move more smoothly through the temporal axis of the RL problem than other methods such as DrQ, RAD, CURL, and SAC-AE.

## 1 Introduction

Learning sample-efficient continuous-control directly from pixels is a challenging task for reinforcement learning (RL) agents. Despite recent advances, modern approaches require hundreds of thousands, if not millions, of agent-environment interactions to attain acceptable performance (Kalashnikov et al., 2018; Jaderberg et al., 2019). In real-world settings, this magnitude of data collection may take a prohibitive amount of time. Therefore, advances in learning efficiency for continuous-control in pixel-based state spaces would significantly accelerate the success of real-world RL applications such as industrial robotics and autonomous vehicles (Dulac-Arnold et al., 2021).

The standard RL loop ties policy optimization and representation learning together with a single supervisory signal. This dual-objective, single-signal learning scheme results in a representation learning bottleneck that ultimately produces agents that understand their environment slowly throughout training (Shelhamer et al., 2017), thereby reducing performance in the early stages of learning. Approaches that have seen the most success in alleviating the bottleneck rely on methods that explicitly encourage the distillation of useful information from the state space.

Recently, data augmentation (Chen et al., 2020a;b; Hénaff et al., 2020) has been studied in the context of RL as a way to improve sample efficiency (Yarats et al., 2021b; Laskin et al., 2020b). Other approaches to improve sample efficiency in RL add auxiliary tasks to the learning process. For example, image reconstruction (Yarats et al., 2020), latent variable models (Lee et al., 2020a; Gelada et al., 2019), variational objectives (Lee et al., 2020b), and contrastive learning (Laskin et al., 2020a; Stooke et al., 2021). While the aforementioned approaches improve performance over base RL algorithms, they do not fully leverage the long-term temporal connection between actions, future states, and rewards. For example, some approaches do not interact with the temporal axis of the RL problem's underlying Markov decision process (MDP) (Yarats et al., 2020; 2021b; Laskin et al.,

2020b), or are only concerned with next-step dynamics (Laskin et al., 2020a; Stooke et al., 2021; Gelada et al., 2019). As we will show with our work, ignoring the long-term temporal axis of the MDP withholds information that can help agents to understand consequences of their environment interactions more quickly. Approaches that do attempt to capture longer-term information either require a significant amount of random exploration and pre-training (Lee et al., 2020a;b), which counteracts the goal of minimizing agent-environment interactions, or condition their predictions of the future on the full-dimensional state space (Lee et al., 2020b), thereby not encouraging latent representations themselves to be predictive of latent representations multiple steps into the future.

In this paper, we revisit the problem of sample-efficient continuous-control in image-based state spaces. We introduce $k$-Step Latent (KSL)[1], a representation learning module that aids RL agents with continuous control on pixels. In contrast to aforementioned prior methods, KSL's auxiliary task directly exploits the RL problem's underlying MDP, requires no pre-training, and encourages latent representations to be predictive of latent representations that are multiple steps into the future. KSL agents learn to create temporally-coherent representations by projecting states into the latent space such that latent representations, when conditioned upon a series of actions, are predictive of latent representations that are multiple steps into the future. KSL's auxiliary task propels agents to new state-of-the-art results on both the *data efficiency* (100k steps) and *asymptotic performance* (500k steps) checkpoints in the popular PlaNet benchmark suite (Hafner et al., 2019).

This paper makes the following contributions. First, KSL augments the learning process of the popular Soft Actor-Critic (Haarnoja et al., 2018a) algorithm to produce new state-of-the-art results in the PlaNet benchmark suite. KSL significantly outperforms current methods such as DrQ (Yarats et al., 2021b) and RAD (Laskin et al., 2020b). Across all six tasks in PlaNet, KSL produces an average improvement over DrQ of 46.9% and 14.5% at the 100k and 500k steps mark, respectively, and an average improvement over RAD of 58.1% and 21.6% at the 100k and 500k steps mark, respectively. Second, we analyze the learned encoders and latent projections that are produced by several methods. Our analyses show that KSL creates representations that are more strongly tied to reward, are less sensitive to perturbation in the state space, and move more smoothly over time than other baseline methods. Also, our analyses show that KSL creates encoders that generalize to unseen tasks better than other baseline methods.

## 2 BACKGROUND

We consider the usual formulation of RL agents that act within a Markov decision process (MDP), defined by the tuple $(\mathcal{S}, \mathcal{A}, P, \mathcal{R}, \gamma)$, where $\mathcal{S}$ is the state space, $\mathcal{A}$ is the action space, $P(s_{t+1}|s_t, a_t)$ is the transition probability distribution, $\mathcal{R} : \mathcal{S} \times \mathcal{A} \to \mathbb{R}$ is a reward function that maps states and actions to a scalar feedback signal, and $\gamma \in [0, 1)$ is a discount factor. The agent's goal is to learn an action-selection policy $\pi(a|s)$ that maximizes the sum of discounted rewards over the time horizon $T$ of the given task: $\arg\max_\pi \mathbb{E}_{a\sim\pi, s\sim P}[\sum_{t=1}^{T} \gamma^t \mathcal{R}(s_t, a_t)]$.

Specifically, we examine learning agents in state spaces comprised of images $o \in \mathcal{O}$. Image-based states are high-dimensional and may give an incomplete description of the environment. For example, incomplete information may arise when the camera's point-of-view shows occluded objects. As such, it is common to stack a series of $n$ consecutive images to help alleviate the problems associated with incomplete information: $s_t = \{o_t, ..., o_{t-n+1}\}$ (Mnih et al., 2015).

### 2.1 SOFT ACTOR-CRITIC

Soft Actor-Critic (SAC) (Haarnoja et al., 2018a;b) is a current state-of-the-art off-policy, model-free RL algorithm for continuous control. SAC learns a state-action value-function critic $Q_\theta$, a stochastic actor $\pi_\phi$, and a temperature $\alpha$ that weighs between maximizing reward and entropy in an augmented RL objective: $\mathbb{E}_{s_t, a_t \sim \pi}[\sum_t \mathcal{R}(s_t, a_t) + \alpha \mathcal{H}(\pi(\cdot|s_t))]$.

SAC's critic is learned by minimizing the squared Bellman error over trajectories $\tau = (s_t, a_t, r_t, s_{t+1})$ sampled from a replay memory $\mathcal{D}$:

$$\mathcal{L}_{critic}(\theta) = \mathbb{E}_{\tau\sim\mathcal{D}}[(Q_\theta(s_t, a_t) - (r_t + \gamma y))^2], \tag{1}$$

---

[1]Code available at: https://github.com/anon-researcher-repo/ksl

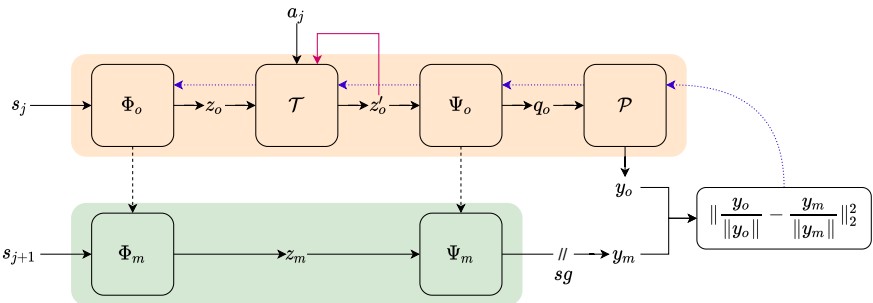

Figure 1: Depiction of KSL's architecture. Online and momentum paths are highlighted in orange and green, respectively. Recurrent operation shown with red arrow. EMA updates shown with dashed arrows. Gradient flow shown with dotted, purple arrows. Stop gradient ($sg$) shown with $//$.

where the temporal-difference target, $y$, is computed by sampling an action from the current policy:

$$y = \mathbb{E}_{a' \sim \pi}[Q_{\bar{\theta}}(s_{t+1}, a') - \alpha \log \pi_\phi(a'|s_{t+1})]. \tag{2}$$

The target critic $Q_{\bar{\theta}}$ is updated as an exponential moving average (EMA): $\bar{\theta} \leftarrow \zeta\bar{\theta} + (1 - \zeta)\theta$ where $\zeta$ controls the "speed" at which $\bar{\theta}$ tracks $\theta$ (e.g., He et al. (2020)). SAC's actor parameterizes a multivariate Gaussian $\mathcal{N}(\mu, \sigma)$ where $\mu$ is a vector of means and $\sigma$ is the diagonal of the covariance matrix. The actor is updated via minimizing :

$$\mathcal{L}_{actor}(\phi) = -\mathbb{E}_{a \sim \pi, \tau \sim \mathcal{D}}[Q_\theta(s_t, a) - \alpha \log \pi_\phi(a|s_t)], \tag{3}$$

and $\alpha$ is simply learned against a target entropy value.

## 3 $k$-STEP LATENT

### 3.1 MOTIVATION

We design KSL to address the representation learning bottleneck of base RL algorithms by providing an auxiliary task that encourages representational consistency of states that are nearby in time. States that are nearby in time are likely to share high levels of mutual information (Mobahi et al., 2009), which implies that the states' representations should as well. KSL contains a recurrent mechanism that encourages representations of states, when conditioned upon actions, to be predictive of states that are multiple steps into the future. Intuitively, we hypothesize that agents who understand the long-term consequences of their actions would perform better than agents who do not.

### 3.2 TRAINING DETAILS

A visual depiction of KSL can be seen in Figure 1 and pseudocode can be found in Appendix B. At each learning step, we uniformly sample $M$ trajectories of length $k$, $\tau = \{(s_1, a_1, ..., a_{k-1}, s_k)_i\}_{i=1}^M$, from the replay memory, where $s_1$ can be any state in the episode. For each state in each trajectory, we apply a random translation augmentation (Laskin et al., 2020b). Using augmentations within KSL encourages representations to be invariant to perturbations over the temporal axis. KSL contains an online and momentum pathway, which are highlighted in Figure 1 as orange and green, respectively. At step $j = 1$ of trajectory[2] $i$, $s_j$ is fed through an online encoder $z_o = \Phi_o(s_j)$ and $s_{j+1}$ is fed through a momentum encoder $z_m = \Phi_m(s_{j+1})$. Next, $z_o$ is concatenated with $a_j$ and fed through a learned transition model $z'_o = \mathcal{T}([z_o|a_j])$. This transition model enables KSL to learn temporal projections from latent representations of state $s_j$ and applied action $a_j$ to a representation of the following state $s_{j+1}$ in the form of $z'_o$. Then, $z'_o$ and $z_m$ are fed through their respective projection modules $q_o = \Psi_o(z'_o)$, $y_m = \Psi_m(z_m)$. Such modules after an encoder have been shown to improve the learned representations of the encoder (Chen et al., 2020a). Finally, $q_o$ is fed through a prediction head $y_o = \mathcal{P}(q_o)$.

---

[2]Note that representations through both pathways are computed for every step and trajectory. We omit batch notation for steps and trajectories in the paragraph for clarity.

KSL's loss for step $j$ is computed as the distance between the $\ell_2$ normalized versions of the final two vectors from both paths: $\|(y_o/\|y_o\|) - (y_m/\|y_m\|)\|_2^2$. This operation is performed in a loop over all $k$ steps per each trajectory in $\tau$, with one notable difference at each step after $j = 1$. Instead of encoding every $s$ along the online path, we recursively feed $z_o'$ from the previous step $j - 1$ into the learned transition model $\mathcal{T}$. This recurrent operation is shown with the red arrow in Figure 1. Therefore, $\Phi_o$ is only used to compute a direct representation of the first state within each trajectory. The minimization of KSL's loss requires $\Phi_o$ to create latent projections such that multi-step consequences can be inferred from any arbitrary starting state using the transition model $\mathcal{T}$. We formulate the loss function as an average over the computed distance for every state and trajectory in a given batch:

$$\mathcal{L}_{KSL} = \frac{1}{Mk} \sum_{i=1}^{M} \sum_{j=1}^{k} \left\| \frac{y_o^{(i,j)}}{\|y_o^{(i,j)}\|} - \frac{y_m^{(i,j)}}{\|y_m^{(i,j)}\|} \right\|_2^2, \tag{4}$$

with $y_p^{(i,j)}$ denoting the final vector of pathway $p$ for trajectory $i$ at step $j$.

We implement a stop gradient to disallow backpropagation from Equation 4 through the momentum pathway. Instead, we update the momentum encoder and momentum projection module as an EMA of their online counterparts. We use the state encoding of the online encoder $\Phi_o$ as input into the critic $Q_\theta$ and actor $\pi_\phi$. While we allow the gradient of $Q_\theta$ to update weights within $\Phi_o$, we implement a stop gradient that disallows signal from the actor's computation graph (Equation 2 and Equation 3) to propagate into $\Phi_o$. Gradient flow is depicted in Figure 1 as purple, dotted lines. $\Phi_m$ is an EMA of $\Phi_o$ and $Q_{\bar{\theta}}$ is an EMA $Q_\theta$, so it is natural for $\Phi_m$ to transform states for input into $Q_{\bar{\theta}}$. See Appendix F for an expanded description of how KSL fits within SAC. We also provide ablations across architecture choices and values of $k$ in Appendix E and Appendix G. We note that KSL is not restricted to SAC and can be applied alongside any RL algorithm for representation learning in tasks with continuous or discrete action spaces.

### 3.3 Optimization Strategy

The weights of $\Phi_o$ are affected by two task-specific losses (Equation 1 and Equation 4). Usually, learning algorithms that attempt to optimize parameters across multiple losses use a weighted linear combination of each loss. However, doing so causes the sharing of inductive biases across each task, which may be an issue if the two tasks compete (Yu et al., 2020; Sener & Koltun, 2018). The representation learning bottleneck of the RL signal may cause disruptions in $\Phi_o$ during optimization of KSL's loss. Additionally, this problem may be exacerbated with optimizers that accumulate gradient statistics (Jean et al., 2018), such as Adam (Kingma & Ba, 2015). Much of the current research in multi-objective learning has focused on methods that actively learn to balance between multiple losses in the context of a single optimizer (Kendall et al., 2018; Groenendijk et al., 2021). Instead, we explicitly separate the optimization of the RL and KSL tasks by sequestering the weights and losses of each task into separate optimizers. This division is not meant to solve the conflicting gradient problem. The division avoids conflation of the Adam optimizer's momentum estimates between the two tasks. As a consequence of our multi-optimizer choice, $\Phi_o$ is granted a higher diversity in gradient-update directions. In our experiments, we see a performance boost due to this separation. See Appendix I for more details.

## 4 Experimental Evaluation

We investigate several facets of KSL and other baseline methods. Most importantly, does KSL result in agents that converge to asymptotic performance in fewer agent-environment interactions than other current methods? We examine KSL's learning progress in environments from the DeepMind Control Suite (DMControl) (Tassa et al., 2018; 2020) (Section 4.1) against several relevant baselines (Section 4.2) and show these results in Section 4.3. Also, we examine the characteristics of encoders and latent representations that KSL and other methods produce. We inspect the relationship between representations and reward, the robustness of encoders to perturbations in the state space, the ability of encoders to generalize to other tasks in the DMControl suite, and the temporal coherence of representations.

## 4.1 DMCONTROL

We examine the learning speed of agents by measuring performance within the popular PlaNet benchmark suite (Hafner et al., 2019). This suite contains environments from DMControl (Tassa et al., 2018) and has been widely used to measure the sample efficiency of RL agents on image-based states. DMControl presents a wide variety of continuous-control tasks, including rigid-body control, control of loose appendages with generated momentum, and bipedal traversal. There are six tasks that range from one-dimensional to six-dimensional action spaces and agents in DMControl are exposed to both dense and sparse reward signals. See Appendix J for more details.

Two checkpoints of interest have emerged in recent literature (e.g., Laskin et al. (2020a)): the 100k steps mark and the 500k steps mark. The former is considered to be the *data efficiency* mark and the latter the *asymptotic performance* mark. The term "steps" refers to the number of environment transitions and not the number of learning updates. Each task has an episode length of 1k steps and has a specific number of action repeats. The agent's chosen action is repeated several times, each of which results in an environment transition. For example, Cheetah, Run has an action repeat of four, which results in 250 actions selections per episode.

## 4.2 BASELINES

We compare KSL against six baseline methods in the data-efficiency experiment: RAD (Laskin et al., 2020b), DrQ (Yarats et al., 2021b), CURL (Laskin et al., 2020a), SAC-AE (Yarats et al., 2020), SAC State, and SAC Pixel. We specifically choose the first four methods because they were (i) state-of-the-art in the PlaNet benchmark suite at one point in time and (ii) all use the exact same encoder and SAC architecture as well as experimental design in PlaNet. Likewise, KSL uses the same encoder and SAC architecture as these four methods to ensure fair comparison. SAC State and SAC Pixel are commonly chosen as experimental control agents. SAC State learns on proprioceptive versions of the state space (e.g., vectors that contain "ground truth" information such as joint angles and velocity). SAC Pixel learns on the pixel version of the state space but with no image augmentation and no representation learning routine. We produce the results for baseline methods using code provided by the original authors. See Appendix D for a comparison between KSL and other methods that do not meet criteria (i) and (ii).

## 4.3 DATA EFFICIENCY EXPERIMENT

During training, all agents perform random actions for the first 1k steps to collect experience tuples for the replay buffer. Each agent type performs one gradient update per action selection by using randomly sampled experience tuples. We perform an agent evaluation every 20k steps. Agent evaluation uses a current snapshot of the learning agent and deterministically samples actions. The agent's final score for each evaluation is the mean of total rewards across 10 episodes. For all methods, we use a batch size of 128. A full table of hyperparameters can be found in Appendix A. We also provide a wall clock time comparison between KSL, RAD, DrQ, CURL, and SAC-AE in Appendix C.

Figure 2 presents the evaluation curves for KSL, RAD, DrQ, CURL, and SAC-AE over 10 seeds. Table 1 highlights the 100k and 500k steps mark as averages $\pm$ one standard deviation, with best mean evaluation returns between image-based methods bolded. We use Welch's t-test to probe for statistical significance between the highest average result and all other results. We highlight that KSL achieves the best average result in all six tasks at both checkpoints of interest and especially outperforms all other methods at the *data efficiency* checkpoint.

## 4.4 ANALYZING LATENT REPRESENTATIONS AND LEARNED ENCODERS

We analyze the latent representations and the learned encoders of all image-based methods from Section 4.2. We collect 10 episodes of state-reward pairs in the Walker, Walk task with a fully-trained agent, which we refer to as the *evaluation* states and rewards. We choose to focus on Walker, Walk as it is one of the more difficult tasks in PlaNet, as judged by the dimension of its action space, the number of action selections per episode, and the difficulty of learning bipedal movement. For the following analyses, we train encoders for each method in Walker, Walk for 100k steps across five seeds, saving the encoders' weights every 5k steps. Specifically, we use the encoders that produce

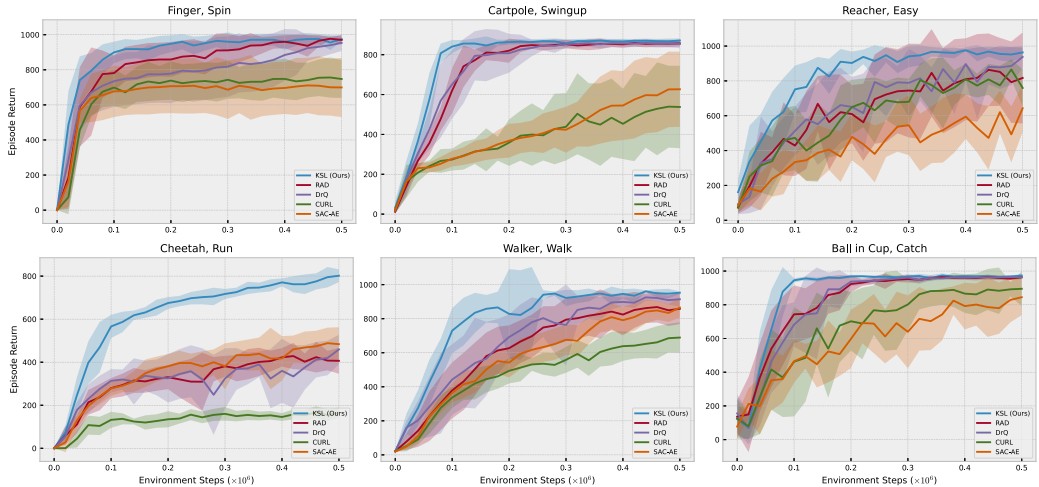

Figure 2: Episodic evaluation returns for agents trained in DMControl. Mean as bold line and $\pm$ one standard deviation as shaded area.

Table 1: Episodic evaluation returns (mean $\pm$ one standard deviation) for PlaNet benchmark. Highest mean results per task shown in bold. * = significant at $p = 0.05$, ** = significant at $p = 0.01$.

| *500k Steps* | KSL | RAD | DrQ | CURL | SAC-AE | SAC State | SAC Pixel |
|---|---|---|---|---|---|---|---|
| Finger, Spin | **$976 \pm 14$** | $970 \pm 28$ | $954 \pm 47$ | $748 \pm 108$ | $699 \pm 169$ | $927 \pm 43$ | $192 \pm 166$ |
| Cartpole, Swingup | **$871 \pm 10$*** | $857 \pm 16$ | $854 \pm 21$ | $538 \pm 207$ | $627 \pm 189$ | $870 \pm 7$ | $419 \pm 40$ |
| Reacher, Easy | **$963 \pm 28$** | $817 \pm 261$ | $938 \pm 63$ | $760 \pm 106$ | $643 \pm 136$ | $975 \pm 5$ | $145 \pm 30$ |
| Cheetah, Run | **$802 \pm 30$*** | $406 \pm 60$ | $459 \pm 69$ | $166 \pm 24$ | $484 \pm 79$ | $772 \pm 60$ | $197 \pm 15$ |
| Walker, Walk | **$953 \pm 8$*** | $858 \pm 90$ | $914 \pm 44$ | $689 \pm 90$ | $865 \pm 59$ | $964 \pm 8$ | $42 \pm 12$ |
| Ball in Cup, Catch | **$973 \pm 9$*** | $963 \pm 11$ | $964 \pm 7$ | $895 \pm 103$ | $845 \pm 106$ | $976 \pm 6$ | $312 \pm 63$ |
| *100k Steps* | | | | | | | |
| Finger, Spin | **$899 \pm 61$*** | $781 \pm 101$ | $735 \pm 146$ | $699 \pm 105$ | $677 \pm 147$ | $672 \pm 76$ | $224 \pm 101$ |
| Cartpole, Swingup | **$841 \pm 33$*** | $620 \pm 114$ | $651 \pm 172$ | $274 \pm 55$ | $278 \pm 39$ | $812 \pm 45$ | $200 \pm 72$ |
| Reacher, Easy | **$751 \pm 137$*** | $429 \pm 175$ | $513 \pm 139$ | $473 \pm 184$ | $334 \pm 55$ | $919 \pm 123$ | $136 \pm 15$ |
| Cheetah, Run | **$566 \pm 54$*** | $279 \pm 47$ | $314 \pm 54$ | $132 \pm 37$ | $278 \pm 35$ | $228 \pm 95$ | $130 \pm 12$ |
| Walker, Walk | **$730 \pm 133$*** | $378 \pm 164$ | $442 \pm 208$ | $337 \pm 62$ | $364 \pm 54$ | $604 \pm 317$ | $127 \pm 24$ |
| Ball in Cup, Catch | **$945 \pm 12$*** | $744 \pm 172$ | $683 \pm 131$ | $462 \pm 324$ | $465 \pm 128$ | $957 \pm 26$ | $97 \pm 27$ |

representations for SAC's critic $Q_\theta$ and actor $\pi_\phi$, which is $\Phi_o$ for KSL. In this section, we refer to these encoders as the *evaluation* encoders.

**Organization Around Reward:** The reward function is the only supervised signal in RL, so it follows that learned representations of the state space should relate to reward. We first observe this relationship qualitatively. Figure 3 shows the first two components of PCA projections of latent representations produced by KSL (top row) and DrQ (bottom row). Here, we focus on the early stages of training to observe emergent behavior in learned representations. The left, middle, and right columns of plots use encoders after 5k, 10k, and 15k environment steps, respectively. Point colors denote the reward. We highlight that KSL's plots show a clear separation between low-reward and high-reward states more quickly than DrQ's plots. By 10k steps, KSL's encoder has grouped the highest-reward states together and by 15k steps, high-reward and low-reward states seem to be linearly separable within the PCA projections. In contrast, DrQ's projections show little sign of reward-based organization by 15k steps. All other baseline methods show the same pattern as DrQ. See Appendix L for PCA plots of other baseline methods. Despite the concrete connection between states and rewards in the context of RL, KSL is the only method that we observe discovering this relationship in the early stages of training.

We also quantify the relationship between learned representations and reward by measuring the mean squared error (MSE) of a linear regression that is fitted to estimate reward from latent projections. Figure 4 displays the mean MSE for all evaluation encoders over all evaluation states. We note

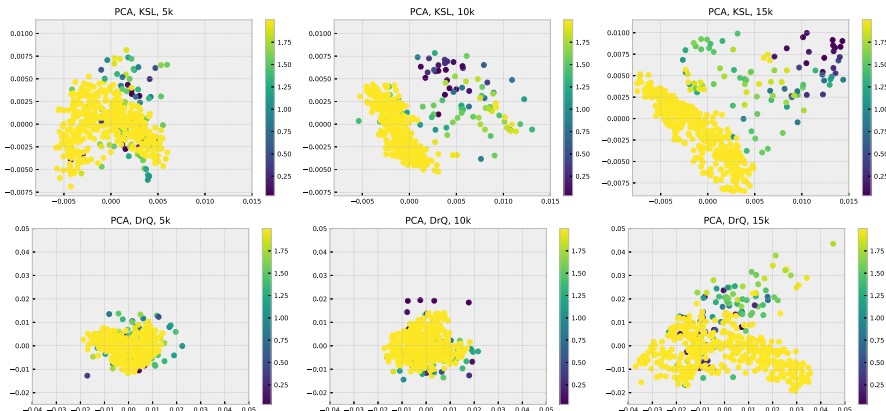

Figure 3: PCA projections of latent representations learned by KSL (top row) and DrQ (bottom row) at 5k (left column), 10k (middle column), and 15k (right column) environment steps. Colors indicate the reward received in the original state.

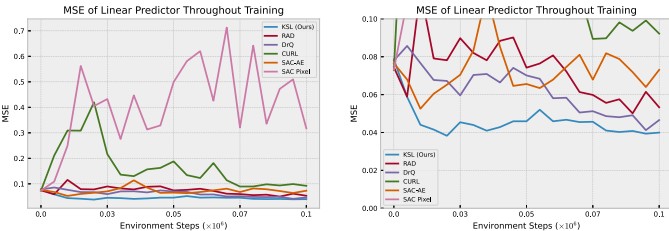

Figure 4: MSE of a linear regression fitted to estimate reward from latent representations. The right plot re-scales the y-axis to show the difference between non-SAC-Pixel methods. Lower is better.

that KSL consistently produces representations that are better linear predictors of reward while also displaying lower variance from one environment step checkpoint to the next than other baseline methods. Also, we note that SAC Pixel produces results that are significantly worse than other baseline methods. This suggests that the RL signal, alone, is not enough to produce representations that are coherently tied to reward.

**Perturbation Invariance:** We measure the invariance of encoders to perturbations in the state space. Specifically, we measure the $\ell_2$ distance between latent representations of several augmented versions of states. We produce three versions of each evaluation state per augmentation using random translation, random erasing, and Gaussian noise ($\mathcal{N}(0, .15)$). Then, we project representations with all evaluation encoders for all augmented states. Figure 5 displays the average $\ell_2$ distance for all evaluation encoders. We note that y-axes are scaled to highlight the curves of baseline methods, which results in SAC Pixel being off-screen. See Appendix M for un-scaled y-axes. These results show the importance of combining image augmentation and auxiliary tasks. Methods that use neither (SAC Pixel) show high sensitivity to all three augmentations. Methods that only use auxiliary tasks (SAC-AE) show a high sensitivity to geometric translations. Methods that use only augmentation (RAD and DrQ) sometimes show instability to the visual perturbance of erasing and Gaussian noise. Methods that use both augmentation and auxiliary tasks (CURL and KSL) show stability across all three augmentations. We highlight that KSL's encoders are generally the most consistent. For example, the change in sensitivity from one evaluation checkpoint to the next is generally much lower for KSL than other baseline methods. See Appendix K for more details.

**Generalization of Encoders:** We test the ability of learned encoders to generalize their projections from one task to another. To this end, we first train each agent in each task for 100k steps five times and save the encoders' weights. Then, we freeze each encoder and train a base SAC agent for 100k steps in every task other than the original training task. Table 2 presents the mean ± one

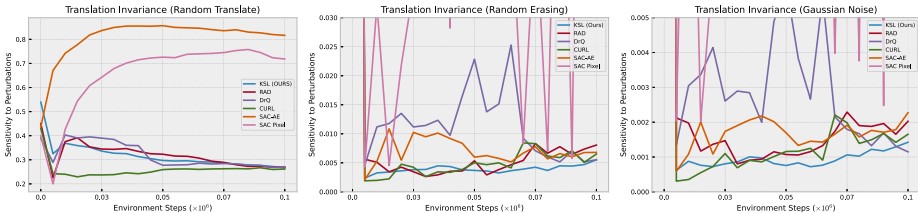

Figure 5: $\ell_2$ distance between latent representations of augmented states. From left-to-right: random translate, random erase, Gaussian noise.

Table 2: Mean $\pm$ one standard deviation of encoder generalization experiment. Best results bolded. * = significant at $p = 0.05$, ** = significant at $p = 0.01$.

| 100k Steps | KSL | RAD | DrQ | CURL | SAC-AE |
|---|---|---|---|---|---|
| Finger, Spin | $705 \pm 179$ | $689 \pm 128$ | $636 \pm 210$ | $\mathbf{722 \pm 108}$ | $693 \pm 127$ |
| Cartpole, Swingup | $\mathbf{325 \pm 64}$ | $258 \pm 62$ | $255 \pm 39$ | $309 \pm 62$ | $240 \pm 29$ |
| Reacher, Easy | $\mathbf{355 \pm 189}$** | $227 \pm 115$ | $206 \pm 150$ | $163 \pm 107$ | $198 \pm 139$ |
| Cheetah, Run | $\mathbf{251 \pm 40}$ | $232 \pm 43$ | $214 \pm 60$ | $235 \pm 34$ | $228 \pm 46$ |
| Walker, Walk | $\mathbf{373 \pm 85}$* | $289 \pm 68$ | $283 \pm 95$ | $320 \pm 71$ | $327 \pm 66$ |
| Ball in Cup, Catch | $\mathbf{566 \pm 219}$* | $329 \pm 216$ | $320 \pm 205$ | $440 \pm 182$ | $297 \pm 200$ |

standard deviation of evaluation returns at 100k steps of training. We highlight that KSL's encoders outperform other methods in five of the six tasks. See Appendix H for full evaluation curves.

**Temporal Coherence:** Control of robotic components, such as those similar to the objects in DM-Control, involves important elements that move smoothly over time, such as joint angles. Ideally, our learned representations should capture this important information from the state space. Therefore, our representations should also move slowly and smoothly across time. We quantify this characteristic by measuring the $\ell_2$ distance between the latent representations of consecutive state pairs in our evaluation set. Figure 6 displays the mean across all evaluation encoders. We highlight that KSL learns representations that move more slowly and consistently than the representations learned by other methods.

## 5 RELATED WORK

### 5.1 REPRESENTATION LEARNING AND REINFORCEMENT LEARNING

KSL builds off of the success of Bootstrap Your Own Latent (BYOL) (Grill et al., 2020) by adapting its architecture to work over the temporal axis of the RL problem's underlying MDP. BYOL is a self-supervised representation learning method for images that does not require negative examples to avoid collapsed representations.

Adding auxiliary tasks to base RL algorithms is a common method to encourage the learning of useful representations. Yarats et al. (2020) introduce SAC-AE, which adds an auxiliary image-reconstruction task with a deterministic auto-encoder (Ghosh et al., 2020) and show that gradients from the actor harm the learning process of encoders. Laskin et al. (2020a) introduce CURL, which augments the RL loop with a projection head akin to contrastive predictive coding (van den Oord et al., 2018) with a momentum encoding procedure (e.g., Chen et al. (2020c)). Alternative tasks have been explored such as aligning the latent- and state-space with bisimulation metrics (Zhang et al., 2021), maximizing mutual temporal-information via learning on hidden activations (Anand et al., 2019), using contrastive learning for task generalization (Agarwal et al., 2021), and using latent dynamics models (Lee et al., 2020a; Hafner et al., 2020; 2021; Gelada et al., 2019; Guo et al., 2020).

Other approaches have relied on using tricks from computer vision, such as image augmentation. RAD (Laskin et al., 2020b) explores how simply augmenting incoming images can help SAC improve without any other alterations. DrQ (Yarats et al., 2021b), published concurrently with RAD,

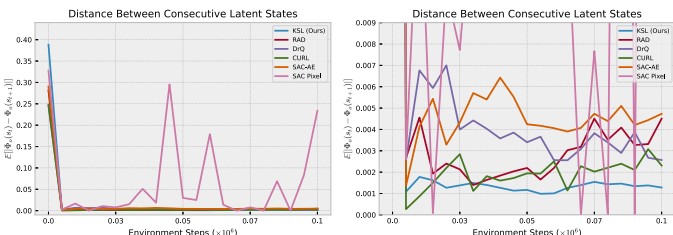

Figure 6: $\ell_2$ distance between consecutive latent states that are produced by the evaluation encoders.

extends RAD to include regularization via averaging Q-function values over several image augmentations during training.

KSL is similar to SPR (Schwarzer et al., 2021) in that both use an architecture similar to BYOL. However, there are several key differences. Unlike SPR, KSL explicitly separates the weight-update process of the RL task and the representation learning task. This separation results in independent tracking of accumulated gradient information within optimizers between tasks. Due to the overlap in parameters between tasks, this separation leads to differences in weight updates. We show in Appendix I that the update direction of the KSL and RL task often conflict throughout training. A high diversity of gradient directions across multiple tasks may be beneficial in the context of non-i.i.d data streams (Aljundi et al., 2019), which is a common scenario in RL. Also, KSL and SPR have significant architectural differences. For one, SPR was designed for discrete-action settings and its architecture relies on one-hot encoded actions. In contrast, KSL's architecture is general enough to be applied in both discrete- and continuous-action domains. Additionally, KSL's transition module operates on vectors, and therefore it contains no convolutional layers. Unlike SPR, this allows KSL to fully share the entire spatial learning process between both the RL and representation learning task. Also, SPR shares more layers than just an encoder between the representation learning and RL task (see note in Figure 2 in Schwarzer et al. (2021)). We refer to this choice as "knowledge sharing". We find that both knowledge sharing (Appendix E) and a convolutional transition model (Appendix G) are detrimental for KSL.

### 5.2 ANALYSIS OF LEARNED REPRESENTATIONS

Most of the focus in RL research is on developing algorithms that provide state-of-the-art performance. Comparatively, less attention has been spent on analyzing the learned representations themselves. Laskin et al. (2020b) qualitatively analyze spatial attention maps (Zagoruyko & Komodakis, 2017) of the learned encoder on various image augmentations. Zhang et al. (2021) provide plots of t-SNE-projected representations to show organization around learned state values. Hafner et al. (2019), and Yarats et al. (2020) observe the relationship between learned representations and the proprioceptive versions of states. We refer the interested reader to the survey work of Bengio et al. (2013), which provides a thorough overview of hypotheses around the question, *"what makes for a good representation, in general?"*.

### 6 CONCLUSION

In this work, we addressed the problem of improving the sample efficiency of reinforcement learning (RL) agents for continuous control in image-based state spaces. To this end, we introduced $k$-Step Latent (KSL), an auxiliary learning task for RL agents. KSL leverages advancements in representation learning and data augmentation to produce state-of-the-art results in the PlaNet benchmark suite. We also analyzed the encoders and latent representations produced by KSL and other baseline methods. These analyses showed that KSL produces encoders that are more robust to perturbations in the state space and generalize to other tasks better than other baseline methods. Also, the analyses showed that KSL's representations are more strongly tied to reward and move more steadily across the temporal axis of the RL problem's underlying MDP. Despite the sample-efficiency gains of KSL, it requires increased wall clock time compared to previous methods. It may be possible to reduce this burden via parallel processing or by reducing the size of KSL's neural networks.

## 7 REPRODUCIBILITY STATEMENT

Please see the footnote on page 2 for a link to an anonymized code repository. Also, see Appendix A for implementation details such as hyperparameters and architecture choices. Finally, see Appendix B for KSL's pseudocode.

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

Table 3: Hyperparameters used to produce paper's main results.

| Hyperparameter | Value |
|---|---|
| Image padding | 4 pixels |
| Initial steps | 1000 |
| Stacked frames | 3 |
| $k$ | 3 |
| Evaluation episodes | 10 |
| Optimizer | Adam |
| $(\beta_1, \beta_2) \rightarrow (\theta, \phi, \Psi, \alpha)$ | (0.9, 0.999) |
| Learning rate $(\theta, \phi, \Psi, \alpha)$ | $2e - 4$ Cheetah, Run |
| | $1e - 3$ otherwise |
| Batch size | 128 |
| Q function EMA $\zeta$ | 0.01 |
| $\Phi$ EMA $\zeta$ | 0.05 |
| Target critic update freq | 2 |
| Actor learning freq | 2 |
| Critic learning freq | 1 |
| Auxiliary learning freq | 1 |
| $dim(z)$ | 50 |
| $\gamma$ | 0.99 |
| Initial $\alpha$ | 0.1 |
| Target $\alpha$ | $- |\mathcal{A}|$ |
| Replay memory capacity | 100,000 |
| Actor log stddev bounds | [-10,2] |

## A  IMPLEMENTATION DETAILS

Each of KSL's encoders $\Phi_o$, $\Phi_m$ are the same as used in SAC-AE, CURL, DrQ, and RAD. Each have four convolutional layers with 32 3×3 kernels. All layers have a stride of one, except for the first, which has a stride of two. The convolutional layers are followed by a single fully-connected layer with 50 hidden units and a layer norm (Ba et al., 2016) operation.

KSL's transition module $\mathcal{T}$ begins with two fully-connected layers with 1024 and 512 hidden units, respectively. These layers are followed by a layer norm operation and a final fully-connected layer that outputs a vector of length 50. Both the momentum and online versions of KSL's projection modules $\Psi_o$, $\Psi_m$ have two fully-connected layers with 512 and 50 hidden units, separated by a layer norm operation. Finally, KSL's prediction head $\mathcal{P}$ is a single 50×50 learnable matrix.

All actor and critic networks are three-layer dense neural networks with 1024 hidden units in the first two layers. All networks use ReLU nonlinearities except for the final output of $\Phi_o$ and $\Phi_m$, which uses tanh.

$\Phi_o$ is used to project states for the critic $Q_\theta$ and the actor $\pi_\phi$. $\Phi_m$ is used to project states for the target critic $Q_{\bar{\theta}}$. We implement stop gradients that disallow propagation from from $\pi_\phi$'s computation graph into $\Phi_o$.

Hyperparameters and algorithm settings can be found in Table 3. $\theta$ refers to all of the parameters in both critics, $\phi$ refers to the actor's parameters, and $\Psi$ refers to all parameters within KSL. The learning frequency refers to the number of action-selections between each learning update. During training, we use uniform random sampling of the replay memory. Initial steps are used to fill the replay memory and are collected by selecting actions randomly.

## B  KSL PSEUDOCODE

Algorithm 1 presents a PyTorch-like pseudocode for KSL. For clarity, we use the same mathematical symbols here as in the main body of the paper. Also, we include PyTorch-like tensor-indexing for clarity. For example, line 3 shows $s[:, 1, :, :, :]$, which means all batches, 1 out of $k$, all color

channels across the stack of images, all pixels in the width, all pixels in the height. There is one minor difference between the notation in Algorithm 1 and the notation in the main body of the paper. To allow for the **for** loop, $z'_o$ is kept as $z_o$ in the pseudocode.

---

**Algorithm 1:** PyTorch-like pseudocode for KSL's loss.

---

1  **Input** : Trajectories $\tau = \{(s_1, a_1, ..., a_{k-1}, s_k)_i\}_{i=1}^{M}$ sampled from replay buffer with augmented states
2  loss $\leftarrow 0$
3  $z_0 \leftarrow \Phi_o(s[:, 1, :, :, :])$
4  **for** $j$ *in* $1...k-1$ **do**
5      $z_m \leftarrow \Phi_m(s[:, j+1, :, :, :])$
6      $z_o \leftarrow \mathcal{T}([z_o|a[:,j]])$
7      $q_o \leftarrow \Psi_o(z_o)$
8      $y_m \leftarrow \Psi_m(z_m)$.detach()
9      $y_o \leftarrow \mathcal{P}(q_o)$
10     loss $\leftarrow$ loss $+ \|\frac{y_o}{\|y_o\|} - \frac{y_m}{\|y_m\|}\|_2^2$.mean()
11 **end**
12 ksl_optimizer.zero_grad()
13 loss.backward()
14 ksl_optimizer.step()

---

## C  WALL CLOCK TIME

To compare wall clock time, we average the total number of seconds to complete 15k episodes of training of Cartpole, Swingup across three seeds. Table 4 shows this average as a value that is normalized to SAC Pixel. All metrics were collected on a machine with a 1080Ti GPU, 8-core AMD Threadripper 1900x @ 3.8GHz, and 128GB of RAM. Codebases for all methods are written in PyTorch.

## D  COMPARISON OF KSL TO PI-SAC AND PROTO-RL

Here, we include a performance comparison between KSL, PI-SAC (Lee et al., 2020b) and Proto-RL (PRL) (Yarats et al., 2021a). PI-SAC uses both a forwards and backwards encoder to learn a compressed representation of image-based states that maximizes the predictive information between representations and future states while minimizing useless information in the state space. We choose to not include this method in Section 4.2 due to significant differences in architecture, hyperparameters, and experiment settings in the PI-SAC paper and papers of methods presented in Section 4.2. For one, PI-SAC's encoder and SAC network architectures are different than those in Section 4.2. Even small differences in network architecture can explain significant differences in performance (Islam et al., 2017; Henderson et al., 2018). Second, PI-SAC uses a significant amount of pre-training to achieve its reported results, while all methods presented in Section 4.2 do not. For example, in Cheetah, Run, Lee et al. (2020b) use 10k pretraining steps, which results in PI-SAC agents receiving the same number of gradient updates that the agents described in Section 4.2 would only receive by the 40k step mark. Thirdly, the hyperparameters used for tasks (e.g., the action repeat) in the PI-SAC paper are not the same as the ones used in previous PlaNet studies. For example, Lee et al. (2020b) use an action repeat of 4 for Cartpole, Swingup, while all methods in Section 4.2 use 8. This results in PI-SAC agents receiving 2x the number of gradient updates for any given environment-steps checkpoint as our baseline methods.

PRL introduces an auxiliary task that encourages intrinsically-motivated agent exploration. PRL agents are trained in two stages. First, the state encoder is pre-trained via a task-agnostic method that attempts to maximize the entropy of the distribution of agent-visited states. Second, the state encoder is frozen and a standard SAC agent is trained on top of the learned representations. We choose to not include this method in Section 4.2 as there are differences in architecture and evaluation procedure between the PRL paper and the papers of methods shown in Section 4.2. For one, PRL's

Table 4: Training time of agent methods, normalized to Pixel SAC.

| Method | Normalized Time |
|--------|-----------------|
| SAC Pixel | 1.0 |
| KSL | 3.9 |
| RAD | 1.2 |
| DrQ | 1.8 |
| CURL | 1.6 |
| SAC-AE | 1.5 |

Table 5: Episodic evaluation returns (mean $\pm$ one standard deviation) for PlaNet benchmark. Highest mean results per task shown in bold. * = significant at $p = 0.05$, ** = significant at $p = 0.01$.

| *500k Steps* | KSL | PI-SAC | PRL |
|--------------|-----|--------|-----|
| Finger, Spin | $\mathbf{976 \pm 14}$ | $936 \pm 84$ | $868 \pm 105$ |
| Cartpole, Swingup | $\mathbf{871 \pm 10}$** | $844 \pm 19$ | $843 \pm 21$ |
| Reacher, Easy | $\mathbf{963 \pm 28}$** | $888 \pm 58$ | $599 \pm 190$ |
| Cheetah, Run | $\mathbf{802 \pm 30}$** | $374 \pm 68$ | $392 \pm 58$ |
| Walker, Walk | $\mathbf{953 \pm 8}$ * | $939 \pm 16$ | $755 \pm 109$ |
| Ball in Cup, Catch | $\mathbf{973 \pm 9}$ ** | $956 \pm 10$ | $949 \pm 25$ |
| *100k Steps* | | | |
| Finger, Spin | $\mathbf{899 \pm 61}$** | $772 \pm 84$ | $658 \pm 88$ |
| Cartpole, Swingup | $\mathbf{841 \pm 33}$** | $722 \pm 94$ | $366 \pm 235$ |
| Reacher, Easy | $\mathbf{751 \pm 137}$** | $337 \pm 143$ | $126 \pm 99$ |
| Cheetah, Run | $\mathbf{566 \pm 54}$** | $227 \pm 55$ | $243 \pm 39$ |
| Walker, Walk | $\mathbf{730 \pm 133}$** | $494 \pm 65$ | $332 \pm 141$ |
| Ball in Cup, Catch | $\mathbf{945 \pm 12}$** | $836 \pm 106$ | $639 \pm 183$ |

encoder produces larger latent vectors than the encoders' of methods in Section 4.2 (50 vs. 128). Second, Yarats et al. (2021a) use an action repeat of two for all tasks and evaluate agents over 1M steps of training. The evaluations in PRL's paper are done in a two-part 500k/500k split. First, 500k steps of auxiliary-task pre-training. Second, 500k steps of RL-task training. We keep this one-to-one ratio in our evaluations. For the 100k steps checkpoint, we train PRL agents in a 50k/50k split. Similarly, we train PRL agents in a 250k/250k split for the 500k steps checkpoint.

We adjust the encoder and SAC networks of PI-SAC and PRL to be congruent with methods in Section 4.2, all hyperparameters to be congruent with Table 3 in Appendix A, and all environment settings to be congruent with Table 7 in Appendix J. Otherwise, we keep PI-SAC and PRL in their original state. We make these adjustments within code provided by the original authors. Table 5 shows the evaluation returns across 10 seeds using the same methodology as described in Section 4.3. Figure 7 shows the evaluation curves. We highlight that KSL agents outperform both PI-SAC and PRL agents in all six tasks for both the 100k and 500k steps mark.

## E    KNOWLEDGE SHARING AND CHOICE OF K

Figure 8 shows potential options for the knowledge sharing (KS) mechanism within KSL. On the far left, we show the Q-function of the critic $Q_\theta$. Although the KS mechanism uses layers from both Q-functions, we only show one in the "Q-function layers" diagram for simplicity. When $l = 1$, only the first layer of the Q-functions are used for KS, and when $l = 2$, the first two layers are used. When $l = 0$, the KS section of KSL's transition module $\mathcal{T}$ is made of a single linear layer that is not part of the Critic's Q-functions.

The choice of $k$ creates a trade-off in terms of algorithm performance and compute expense. A larger $k$ gives KSL access to deeper levels of temporal information. However, a larger $k$ results in a larger amount of compute, both in terms of data sampling and the recurrent loop within KSL. With every incremental $k$, KSL needs to sample and perform the translation augmentation on *batch*

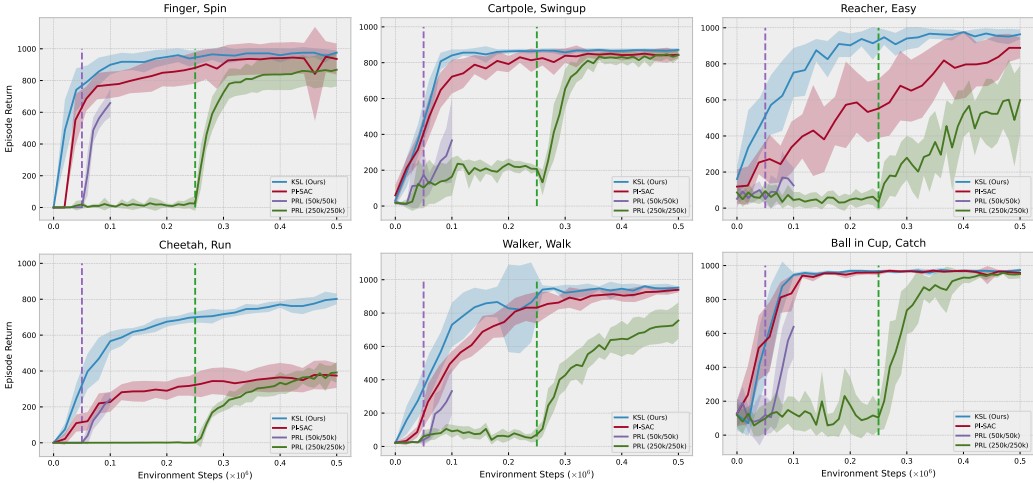

Figure 7: Episodic evaluation returns for KSL, PI-SAC, and PRL agents trained in PlaNet benchmark suite. Mean as bold line and $\pm$ one standard deviation as shaded area. Purple and green dotted lines indicate when 50k/50k PRL and 250k/250k PRL, respectively, begin to receive task-specific rewards.

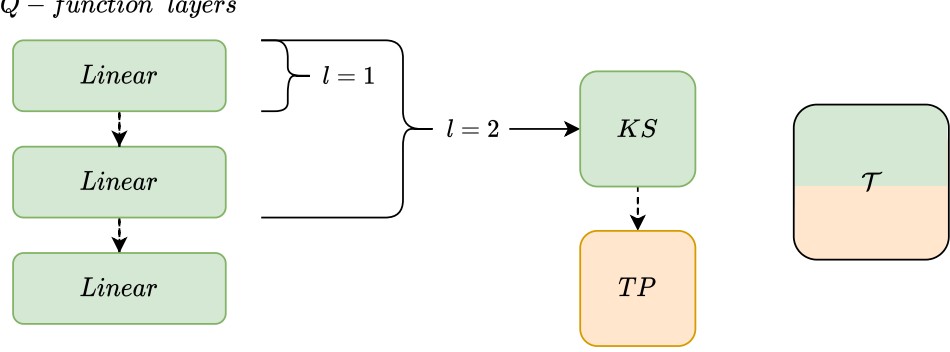

Figure 8: Depiction of options for knowledge sharing. Critic's Q-functions shown on far left. KSL's transition module $\mathcal{T}$ is shown on far right with its two components: knowlege sharing (KS) and transition predictor (TP). Data-flow is shown with dashed lines.

Table 6: Ablation experiment: 100k steps (left) and 500k steps (right).

|  | $l = 0$ | $l = 1$ | $l = 2$ |
|---|---|---|---|
| $k = 1$ | $493 \pm 33$ | $493 \pm 59$ | $512 \pm 40$ |
| $k = 3$ | $\mathbf{575 \pm 58}$ | $538 \pm 13$ | $479 \pm 73$ |
| $k = 5$ | $535 \pm 14$ | $536 \pm 37$ | $517 \pm 36$ |

|  | $l = 0$ | $l = 1$ | $l = 2$ |
|---|---|---|---|
| $k = 1$ | $725 \pm 15$ | $728 \pm 43$ | $711 \pm 31$ |
| $k = 3$ | $\mathbf{798 \pm 30}$ | $782 \pm 36$ | $703 \pm 138$ |
| $k = 5$ | $763 \pm 40$ | $775 \pm 32$ | $712 \pm 35$ |

*size* more states from the replay memory. Then, each additional state needs to perform several additional passes through neural network modules as well as compute a loss, therefore leading to an exponential growth in compute requirements.

We perform a full grid-sweep across all combinations of $k \in \{1, 3, 5\}$ and $l \in \{0, 1, 2\}$ in the task of Cheetah, Run. Figure 9, below, shows the mean (bold line) and one standard deviation (shaded area) across five random seeds. Table 6 summarizes these results for the 100k and 500k checkpoints.

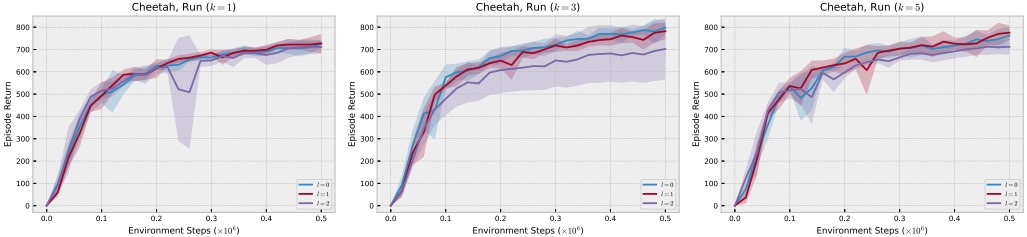

Figure 9: Grid search performance for KSL for $k \in \{1, 3, 5\}$ (left to right) and $l \in \{0, 1, 2\}$. Mean performance shown in bold and $\pm$ one standard deviation with shaded area.

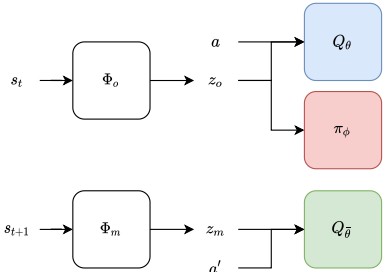

Figure 10: Depiction of how KSL fits within the greater SAC algorithm.

These results show that knowledge sharing provides no benefit and may actually harm the RL learning process. In general, when $k > 1$, $l > 0$ leads to lower levels of performance with higher variance between runs. Also, the results show that there is little reason to go beyond $k = 3$. As such, the main results in the paper use $l = 0$ and $k = 3$.

## F  HOW KSL FITS WITHIN SAC

Figure 10 shows how KSL fits within the greater SAC algorithm. $\Phi_o$ takes in current states $s_t$ and creates latent projections $z_o$ for both the critic $Q_\theta$ (blue) and the actor $\pi_\phi$ (red). $\Phi_m$ takes in future states $s_{t+1}$ and creates latent projections $z_m$ for the target critic $Q_{\bar{\theta}}$ (green).

## G  CONVOLUTIONAL TRANSITION MODEL

We test the performance of a convolutional transition model $\mathcal{T}$ within KSL. To make a convolutional $\mathcal{T}$ fit within KSL, we adjust its architecture. First, we remove the final dense layer from the encoders $\Phi$. $\mathcal{T}$ is then made of two convolutional layers with 32 and 16 filters, respectively, and both have a $3 \times 3$ kernel with a stride of two. Following the convolutional layers is a single dense layer with 1024 $+dim(\mathcal{A})$ hidden units that outputs a vector of length 50, followed by a layer norm operation. Action vectors are concatenated to the flattened feature maps from the transition model's convolutional layers before being fed through its dense layer. Both the critics and the actor have two convolutional layers added to their networks, all with the same architecture as described here for $\mathcal{T}$. Additionally, they contain one dense layer with 1024 hidden units that outputs a vector of length 50. The remaining architectures of $\Phi_m$, $\Phi_o$, $\mathcal{P}$, $Q_\theta$, $Q_{\bar{\theta}}$, and $\pi_\phi$ remain unchanged. The additional layers in the critic target are updated as an EMA of their counterparts in the critic.

Figure 11 shows the evaluation results for both KSL with (five seeds) and without (ten seeds) a convolutional $\mathcal{T}$. We highlight that the version of KSL with a convolutional $\mathcal{T}$ performs significantly worse than the version of KSL with a fully-dense $\mathcal{T}$. Also, we notice a significantly higher probability of representational collapse in the version of KSL with a convolutional $\mathcal{T}$. As a result, we use the version of KSL with a fully-dense $\mathcal{T}$ for all other results in this paper.

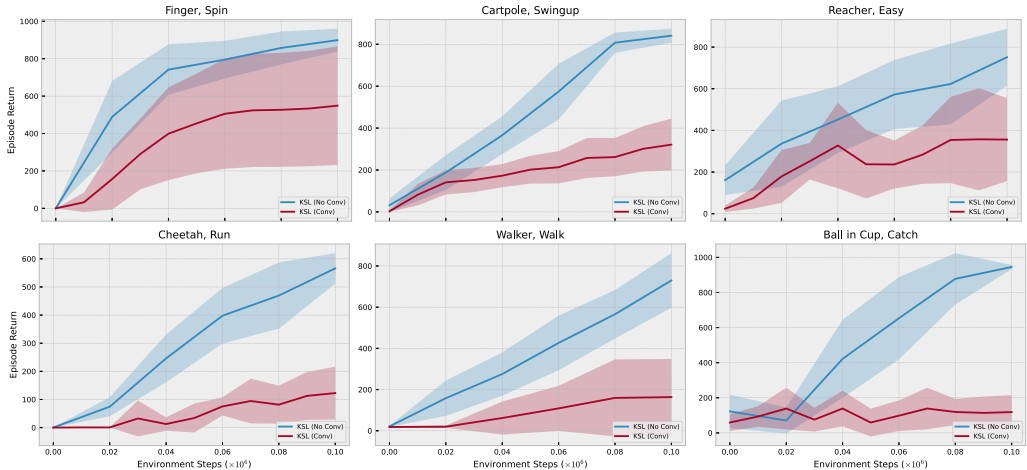

Figure 11: Mean $\pm$ one standard deviation of evaluation returns for convolutional (conv) and fully-dense (no conv) transition model.

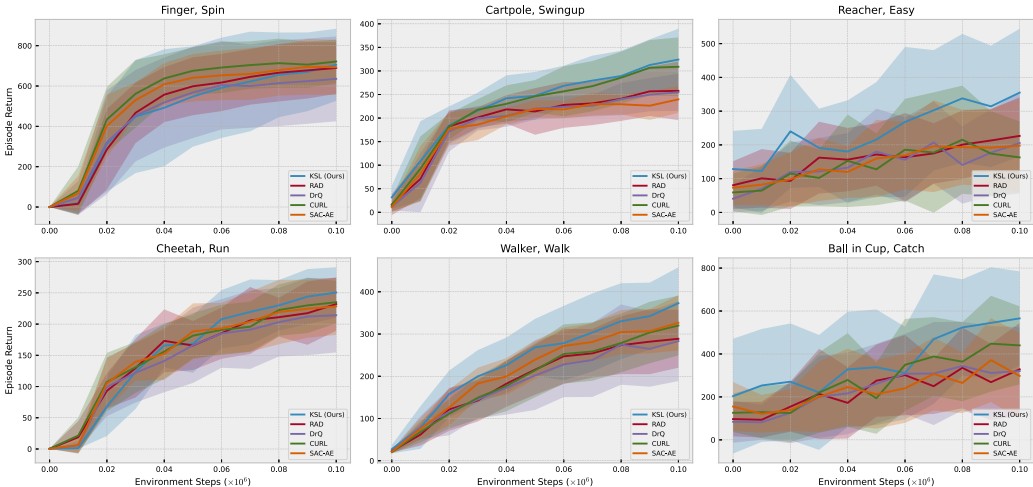

Figure 12: Evaluation curves for the encoder-generalization experiment.

## H  GENERALIZATION LEARNING CURVES

Figure 12 shows the learning curves for the generalization experiment.

## I  OPTIMIZERS VS. TASKS

The Adam optimizer (Kingma & Ba, 2015) performs optimization updates by using exponentially-decayed first moment $\hat{m}$ and second moment $\hat{v}$ estimates of historical gradients. Adam updates parameters $\theta$ at timestep $t$ as

$$\theta_{t+1} \leftarrow \theta_t - \alpha \overbrace{\frac{\hat{m}_t}{\sqrt{\hat{v}_t} + \epsilon}}^{\text{Update direction}} ,$$

(5)

where $\epsilon$ is a small constant.

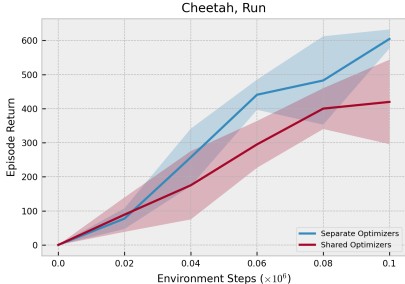
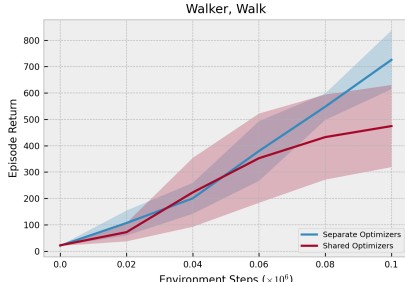

Figure 13: Performance of agents on Cheetah, Run (left) and Walker, Walk (right) for scenario (i) (red) and (ii) (blue). Plots depict averages (bold line) and one standard deviation (shaded area) over five runs.

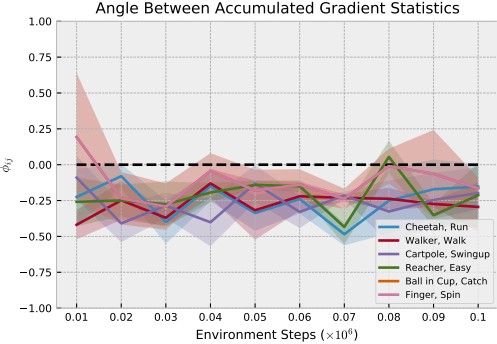

Figure 14: $\phi_{ij}$ for for all tasks in the PlaNet benchmark suite throughout training.

In KSL's formulation, the RL task and the representation learning task both send feedback from their loss functions into $\Phi_o$. Two unique situations can arise with the optimization process in KSL: (i) a single optimizer handling updates from both tasks or (ii) separate optimizers handling updates for either task. The differences in optimizer-held gradient statistics results in different optimization steps for $\Phi_o$ in either scenario. Using the same optimizer for both tasks may result in updates that conflate the accumulated gradient statistics of $\Phi_o$ between tasks.

To confirm this, we perform two evaluations. First, we measure the learning performance of KSL agents for scenario (i) and (ii) in both Cheetah, Run and Walker, Walk. Figure 13, below, shows the average and one standard deviation across five seeds. We highlight that KSL agents that use separate optimizers for either task result, on average, in better learning performance. Also, we note that these results are statistically significant at $p = 0.05$.

Second, we query $\hat{m}$ and $\hat{v}$ from either optimizer in case (ii) throughout training for all tasks in the PlaNet benchmark suite. We extract the accumulated statistics for $\Phi_o$ and measure the angle $\phi_{ij} = \langle g_i, g_j \rangle / \|g_i\| \|g_j\|$ where $g_i$ and $g_j$ are the "update direction", as denoted in Equation 5, for the RL- and KSL-task optimizers, respectively. When $\phi_{ij} < 0$, task-specific updates are considered to be conflicting. When gradients are conflicting, updates move weights in "opposing" directions. Figure 14 depicts $\phi_{ij}$ for all tasks in PlaNet over three runs of training. We highlight that, on average, $\phi_{ij} < 0$ for all tasks. This implies that maintaining a combined $\hat{m}$ and $\hat{v}$ for both the RL and KSL task conflates update information from both tasks.

The RL task is unlike other optimization problems in that its sequential nature and non-i.i.d data create a moving target. Given that KSL's auxiliary task improves the performance on the RL task, we hypothesize that the pull from KSL's conflicting update helps to regularize learning of the RL task. Also, the independent tracking of gradient statistics helps to increase the diversity of gradients, which has been shown to improve learning in streams of non-i.i.d data (Aljundi et al., 2019).

Table 7: Dimensions of action spaces, action repeat values, and reward function type for all six environments in the PlaNet benchmark suite.

| Environment, Task | $dim(\mathcal{A})$ | Action Repeat | Reward Type |
|---|---|---|---|
| Finger, spin | 2 | 2 | Dense |
| Cartpole, swingup | 1 | 8 | Dense |
| Reacher, easy | 2 | 4 | Sparse |
| Cheetah, run | 6 | 4 | Dense |
| Walker, walk | 6 | 2 | Dense |
| Ball in Cup, catch | 2 | 4 | Sparse |

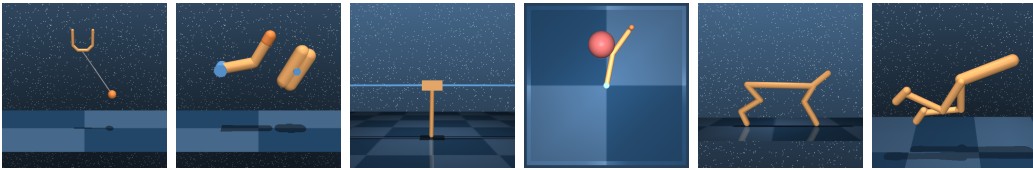

Figure 15: Depiction of the six tasks tasks in the PlaNet benchmark suite.

Alternatives to our multi-optimizer choice may include altering the gradients of the tasks before they are combined and passed to the optimizer (e.g., Yu et al. (2020)), changing the architecture of $\Phi_o$ to include task-specific attention (Liu et al., 2019), or applying additional learning procedures that control the learning schedule between tasks (e.g., Xu et al. (2019)).

## J    ENVIRONMENT DETAILS

Table 7 describes the dimensions of the action space, the number of action repeats, and the type of reward (dense vs. sparse) for each task in the PlaNet benchmark suite. Figure 15 depicts the six tasks within the PlaNet benchmark suite.

## K    EXTENDED ENCODER SENSITIVITY

We measure the change in sensitivity to perturbations as the absolute difference between one encoder checkpoint to the next. A lower value is desireable, as it indicates smooth changes in latent representations throughout the training process. Smooth changes in latent representations results in less variability in input for the downstream RL training task. Best results are bolded. Table 8 shows this metric.

## L    EXTENDED PCA PLOTS

Here we provide PCA plots for all methods not shown in the main body of the paper. Figure 16, Figure 17, Figure 18, Figure 19 show plots for SAC Pixel, RAD, CURL, and SAC-AE, respectively.

## M    EXTENDED AUGMENTATION PLOTS

Figure 20 shows plots for the random erasing augmentation and Figure 21 shows plots for the Gaussian noise augmentation.

Table 8: Change in sensitivity from one encoder checkpoint to the next.

| Method | Translation | Erasing | Flipping |
|--------|-------------|---------|----------|
| KSL | .008 | **.0004** | **.001** |
| RAD | .016 | .001 | .006 |
| DrQ | .015 | .004 | .018 |
| CURL | **.004** | .001 | .005 |
| SAC-AE | .012 | .002 | .004 |
| SAC Pixel | .032 | .088 | .452 |

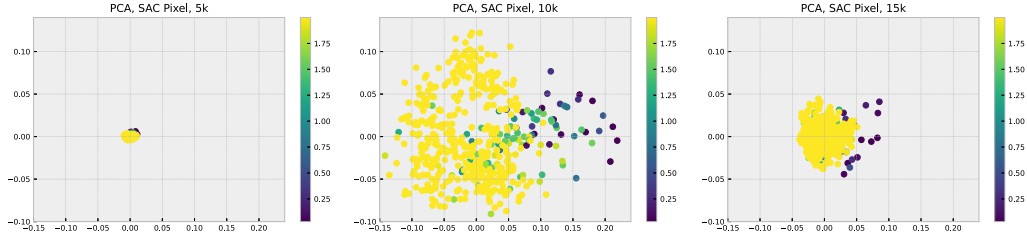

Figure 16: First two principal components of SAC Pixel.

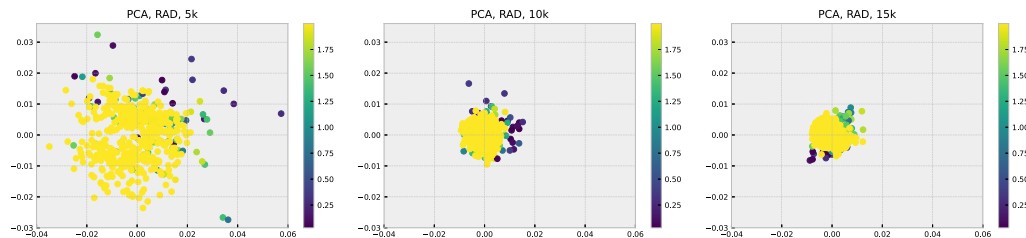

Figure 17: First two principal components of RAD.

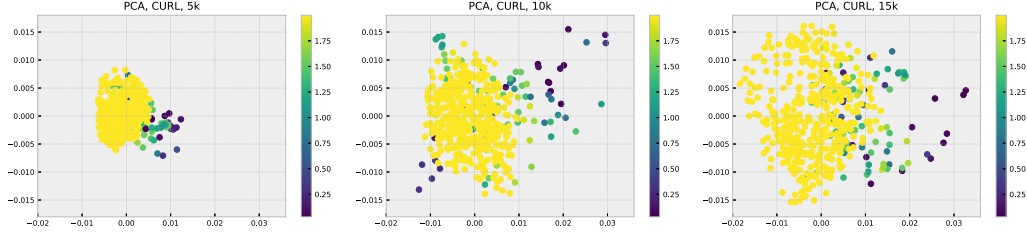

Figure 18: First two principal components of CURL.

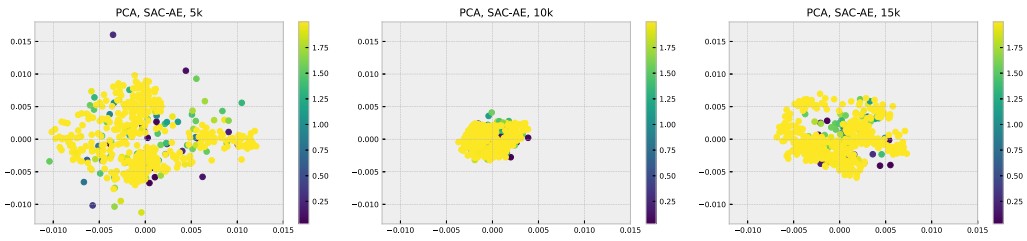

Figure 19: First two principal components of SAC-AE.

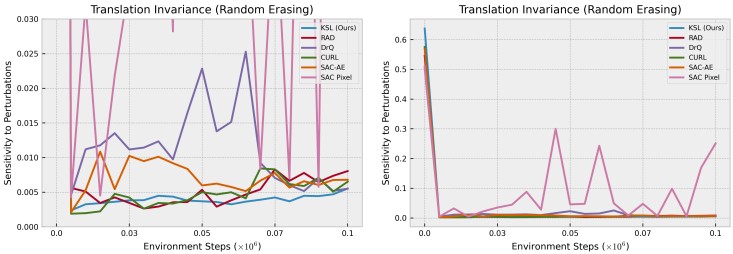

Figure 20: Random erase augmentation.

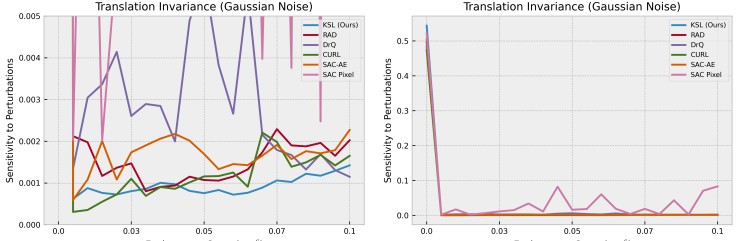

Figure 21: Gaussian noise augmentation.

