# OpenReview forum: "Learning Temporally-Consistent Representations for Data-Efficient Reinforcement Learning"
_ICLR.cc/2022/Conference — ICLR 2022 Submitted_

### Official Review · Reviewer_Qru8 · 2021-11-02

**Correctness:** 3
**Technical Novelty And Significance:** 2
**Empirical Novelty And Significance:** 3
**Recommendation:** 5
**Confidence:** 4

**Main Review:**

Pros:

 - The paper is well-written and easy to comprehend;

 - The empirical evaluation on dm-control suite indeed shows that KSL yields state-of-the-art results on the six presented tasks;

 - The choice of using independent optimisers for training the representation learning module given the signals from the predictive latent supervision (Eq. 4) and critic training (Eq. 1) respectively raises a good point for representation learning in RL with auxiliary tasks, from multi-task learning perspective. Such inductive bias is additionally substantiated with empirical comparisons.

Concerns:

 - The main evaluation on the overall performance is rather limited, it might worth showing more (especially the middle/hard tasks as defined in Yarats, et al. 2021);

 - The KSL model combines many existing techniques in representation learning for RL, such as image augmentation (Laskin, et al. 2020) and multi-step latent predictive supervision auxiliary task (Schwarzer, et al. 2021), leading to limited novelty of the proposed KSL model.

 - The multi-step latent predictive supervision objective for representation learning in KSL is highly similar to the SPR model (Schwarzer, et al. 2021), it seems like the authors acknowledge the similarity hence spent a paragraph discussing the difference, but discussions are mainly based on the architectural/training differences. From this perspective, KSL appears as an adaptation of the SPR model to the continuous control tasks (despite the admirable engineering efforts). The difference argument could be made convincing given that the authors could provide further clarifications of the differences between KSL and SPR, from an algorithmic perspective, and provide further empirical comparisons between the two agents (e.g., the authors state that "KSL’s architecture is general enough to be applied in both discrete- and continuous-action domains", hence it would be interesting to see an adaptation of KSL to discrete control and test it on atari benchmarks, and compare with the SPR results).

 - The arguments of the improved representation learning of KSL in terms of "Permutation Invariance" and "Temporal Coherence" seems poorly justified. The reported figures do not show significant or consistent improvement of the KSL over the baseline methods, only by largely zooming in the y-axis could one observe some differences. I doubt if such minuscule differences are consequential in the overall learning.

 - I think using the momentum encoder to provide input to the target Q-network in SAC critic-training is an interesting choice, but lacks further (theoretical) justification, i.e., why would this be better than simply using the online encoder as the inputs to the target Q-network, is it possibly because of the consistent temporal lags?

Minor Points:

 - The main empirical evaluations of the learned representations is based on the walker-walk task, which is a simple task with dense reward structure(Yarats, et al. 2021). It would be more interesting to see how the learned representations are indicative of the reward structure in the sparse reward tasks, such that Cartpole-Swingup.

 - KSL is motivated by the inductive bias that "States that are nearby in time are likely to share high levels of mutual information", another similar work that utilises multi-step action-dependent latent predictions by Whitney, et al. (2019) is based on the inductive bias that the similarities between the embeddings for the states and/or action sequences should be based on their successor outcomes (e.g., successor representations). It would be nice to see some discussions on the relationship between the two seemingly independent inductive biases.


**Summary Of The Paper:**

Summary:
The authors introduced k-Step Latent (KSL), a representation learning method for visual-based continuous control tasks. KSL utilises multi-step latent action-dependent predictive supervision for training the representation. The empirical evaluations are based on the dm-control suite benchmarks, KSL demonstrates improved sample efficiency (100k evaluation) and asymptotic performance (500k evaluation) across the six tasks presented, comparing to the baseline algorithms (mainly based on image augmentation). The authors further empirically examined the properties of the learned representations, and showed that the trained encoder quickly learns to be representative of the reward structure. The authors also argue that KSL supports more robust representation learning and stronger generalisability.


**Summary Of The Review:**

Scores:

I suggest marginal rejection (5/10). KSL indeed show state-of-the-art performance on the presented tasks and I like the way the authors assessed the quality of representation learning. However, KSL seems like a combination of existing methods, but lacks comprehensive empirical evaluation in that sense. Moreover, the high similarity with SPR is concerning without further clarification and empirical justification. Some arguments of the improvement learned representation is over-stated.

---

> ### Author Response · Authors · 2021-11-22
> **Comment for Reviewer Qru8**
>
> We thank Reviewer Qru8 for their time and suggestions.
>
> - $\textbf{Limited evaluation:}$ We agree that more performance evaluations would improve the paper. We thank Reviewer Qru8 for pointing us to two specific sets of environments.
>
> - $\textbf{Comparisons with SPR:}$  We thank Reviewer Qru8 for providing several suggestions on how to further differentiate KSL and SPR. In future versions of the paper, we will make sure to further clarify any differences.
>
> - $\textbf{Further benchmarks within Atari:}$ We agree that including  performance evaluations within the Atari benchmark suite would enhance the paper. However, the compute requirements for Atari make it unlikely that we would be able to gather a sufficient number of seeds in such a short amount of time. Instead, we will focus on gathering this data in a future version of the paper.
>
> - $\textbf{Permutation Invariance and Temporal Coherence:}$ We agree that the difference between our collected metrics appears small. However, it may be possible that such a magnitude of difference could affect the learning process. Perhaps the representation evaluation section could be improved if we also quantified the downstream effects of such differences.

---

### Official Review · Reviewer_38iT · 2021-11-02

**Correctness:** 3
**Technical Novelty And Significance:** 2
**Empirical Novelty And Significance:** 2
**Recommendation:** 3
**Confidence:** 4

**Main Review:**

**Clarity.**
For the most part, the writing is clear. One minor concern is that the description of KSL (sec 3.2) is difficult to understand, and could be made much more clear with better choices of notation, including the pseudocode in the main text, and an explanation of the algorithm that follows the pseudocode.

That said, the main issue with the clarity and writing of the paper is that the contribution is not made clear both in the paper nor in the experimental analysis. Is the paper about using k-step latents plus an auxiliary loss tying these to observations? Or is the paper about the specific form this auxiliary loss takes? Most of the writing and the naming (KSL) imply the former but the evaluation speaks to the latter.

**Novelty and significance.**
Clarity aside, the paper does not propose a sufficiently novel contribution for acceptance. There is a very large body of work in model-based reinforcement learning that uses k-step model-based predictions and corresponding losses to improve data efficiency and performance, and reduce model rollout errors. Some of this is even cited in this paper, but others that are not cited include Recurrent environment simulators by Chiappa et al 2017, TreeQN by Farquhar et al 2018, MuZero by Schrittwieser et al 2019, and Muesli by Hessel et al 2021. These methods all use k-step latents with a loss tying them to (encodings of) observations.

The specific form of the auxiliary loss here, adapted from BYOL (which is not even mentioned until the related work for some reason), is very similar to that of SPR, which is cited by the paper. There are some minor differences but these mainly seem like implementation details and since there are no empirical comparisons I have to assume this is the case.

Further, the experiments themselves are extremely limited. Evaluating only on 6 tasks from the DM control suite is not enough to show that this is a compelling and useful contribution, especially given the high overlap with prior work. The additional analysis of the learned representations are nice, but are not enough without showing the strength of the proposed approach, or else doing a much more thorough analysis. Finally, I’d like to see ablations of the components and choices made for the proposed method. Why have both \Psi_o and P? Is using the EMA for the momentum pathway the best choice? Is a normalized L2 loss the best choice?


**Summary Of The Paper:**

This paper proposes a representation-learning method (k-step latent, or KSL) that uses a self-supervised auxiliary loss between recurrently-predicted action-conditioned representations of the state space and non-recurrently predicted target representations, in the style of BYOL. The method is trained using two separate optimizers on different parts of the models, to avoid interference in the statistics maintained by the optimizers. Results at 100k and 500k steps on 6 tasks from the DM control suite (from pixels) compared to methods using alternative self-supervised auxiliary losses show that the proposed method improves data efficiency. Analysis of the learned latent representations shows that those from KSL produce more robust encoders and are more consistent with the underlying MDP.

**Summary Of The Review:**

Overall, this paper lacks sufficient novelty for acceptance. It recombines existing techniques in a slightly different way than previously and shows improvements on a very small and narrow set of environments without comparing to the most relevant related work.

---

> ### Author Response · Authors · 2021-11-22
> **Comment for Reviewer 38iT**
>
> We thank Reviewer 38iT for their time and suggestions.
>
> - $\textbf{Clarity of writing:}$ To clarify, the auxiliary loss attempts to tie together latent representations over the time axis. Also, we will try to change the notation in 3.2 to make the description more clear, per Reviewer 38iT's suggestion.
>
> - $\textbf{Sufficient novelty when compared to model-based RL:}$ We thank Reviewer 38iT for providing several examples of model-based RL methods that may be related. We will make sure to include mention of methods that have overlap with our method.
>
> - $\textbf{Limited experiments:}$ We agree that adding additional performance evaluations would help strengthen the paper. Some of the other reviewers have suggested additional evaluation environments.
>
> - $\textbf{Additional ablations:}$ We provided a number of ablations in the Appendix of the paper for both architecture and hyperparameter choices. However, we did not perform ablations for the three choices that Reviewer 38iT mentioned. It may be worth exploring these avenues, and we thank Reviewer 38iT for suggesting them.

---

### Official Review · Reviewer_TKuY · 2021-11-03

**Correctness:** 3
**Technical Novelty And Significance:** 3
**Empirical Novelty And Significance:** 3
**Recommendation:** 5
**Confidence:** 4

**Main Review:**

Strenghts:
* Simple approach
* Clearly written
* Representation learning for better generalization or sample efficiency is an important topic in RL
* Positive experimental results

Weaknesses:
* My main worry with this application of BYOL to RL is that the introduction of the transition model T changes the 'support' of z_m away from the support of z_o. In other words, while psi_o expects an output of T, psi_m gets the direct output of phi_m for which it was not trained and which might be entirely different from the output of T. Based on the experiments, this still seems to do something useful, but I would argue that psi_m should be seen more as random mapping than as the slow moving state-encoding. In any case, I think this out-of-distribution problem for psi_m should be addressed in the paper.
* For Figure 3: Why not use t-SNI instead of only the first two dimensions of PCA? In particular, while it's not wrong to say that "DrQ's projections show little sign of reward-based orgnaization by 15k steps", that is slightly misleading as it doesn't say anything about the latent representation as we're only looking at 2 principal axes.

Additional Questions:
* How is translation augmentation applied?
* Nit: How would the results change when removing the sg before the policy?

**Summary Of The Paper:**

The paper applied "Bootstrap your own latent" (BYOL) to the case of RL by introducing an additional learned transition model and show that this can improve sample efficiency.

**Summary Of The Review:**

An interesting direct application of BYOL to RL. However, the necessity to include action-conditioned transition models in RL raises additional complications compared to BYOL which have not yet been addressed (or discussed) and I believe these should be included in the paper before publication.

---

> ### Author Response · Authors · 2021-11-22
> **Comment for Reviewer TKuY**
>
> We thank Reviewer TKuY for their time and suggestions.
>
> - $\textbf{Introduction of T:}$ Reviewer TKuY raises a very interesting point. As far as we are aware, there is little literature that covers the theoretical underpinnings of how BYOL learns (if Reviewer TKuY knows of such papers, we would appreciate being pointed to them!). As such, it may be difficult to fully compare how the addition of T changes the nature of BYOL's learning process.
>
> - $\textbf{Usage of PCA:}$ We chose to not use t-SNE (or any of its derivatives) as these algorithms learn their own non-linear mapping that discovers structure and are dependent on hyperparameters. Instead, we wanted to determine if patterns existed without such a mapping. Also, we chose the first two components to create clear visuals.

---

### Official Review · Reviewer_pTps · 2021-11-03

**Correctness:** 3
**Technical Novelty And Significance:** 2
**Empirical Novelty And Significance:** 3
**Recommendation:** 6
**Confidence:** 3

**Main Review:**

The paper addresses a very important problem in RL: data efficiency. The author's perspective on leveraging long-term temporal connection is not exactly novel (see Self-Predictive Representation [Schwarzer et al. 2021], Successor Features [Kulkarni, et al. 2016, Barreto et al. 2017]), but the specific method introduced seems to be novel, to the best of my knowledge.

On the method:
The motivation is clearly stated and makes sense to me. I would have like to see discussion on how this differ/relates to successor features, when learned jointly or separately.

A claimed "that learned representations of the state space should relate to reward" is not really justified: there are data-efficient method that disentangle the reward from the representation. It also seems to contradict another desired property: generalisation from one task to another.

On the experiment:
The experiments are extensive and the method is compared against sensible baselines. The results with respect to data-efficiency are promising.


minor comments:
- the font is very small on most figures axes
- figures 5.2, 5.3 and 6 would make more sense with a different y axis scale.

**Summary Of The Paper:**

The paper tackles the problem of sample inefficiency in continuous control, by noting that standard RL methods deal with both policy optimisation and representation learning jointly with a single supervisory signal, namely the reward. Consequently, authors propose to leverage long-term temporal connections between actions in the representation learning, and introduce k-Step Latent (KSL), a representation learning module for learning temporally consistent representations of the state space. Authors show that KSL improves previous state-of-the-art methods in PlaNet benchmark suite and provide some analysis of the representations learned by KSL.

**Summary Of The Review:**

The main idea behind the paper is not novel, but the implementation is, and the results are promising. Some claims are not founded and somehow contradictory, and some related works are missing. But overall an interesting contribution!

---

> ### Author Response · Authors · 2021-11-22
> **Comment for Reviewer pTps**
>
> We thank Reviewer pTps for their time and suggestions.
>
> - $\textbf{Successor Representations/Features (SR):}$ Thank you for pointing out these examples of SR literature. We agree that these works follow similar goals and partly follow similar architectures and objectives to our work. Therefore, we agree that adding further discussion and ablations to compare/contrast KSL with prior SR algorithms makes sense to highlight the novelty and differences of KSL with respect to related work.
>
> - $\textbf{Claim on representations relating to reward:}$ The main intuition here is the reward function $\mathcal{R} : \mathcal{S}  \times  \mathcal{A} \rightarrow \mathbb{R}$, which provides a reward based both on states and actions. Perhaps our phrasing is too strongly worded and should read more like "...encouraging a relationship between the learned representations and reward should improve performance." However, your point regarding representations reflecting rewards and whether this is desirable when generalization is one of the goals of our representations is sensible. We want to point out that KSL's representations might correspond to reward functions across multiple tasks due to their temporal component. In particular, in the DMControl environments, reward functions are similar across most tasks, but we believe this might require further investigation.

---

### Official Review · Reviewer_J6YX · 2021-11-06

**Correctness:** 2
**Technical Novelty And Significance:** 2
**Empirical Novelty And Significance:** 2
**Recommendation:** 3
**Confidence:** 4

**Main Review:**

Data-efficiency of RL algorithms is an important research area, and this paper explores how auxiliary self-supervised learning can improve data-efficiency in continuous control domains. The paper is very rich with experimental details, the implementation choices are carefully ablated and the paper is overall well written and explained.

### Similarities with SPR
------
My main issue with this paper is the fact that it positions the K-step latent objective as a “new representation learning method”, whereas in fact the K-step latent is exactly the representation learning method used in SPR [1]. Throughout the abstract, intro and the methods section, the paper positions KSL as a new representation method: but in practice it’s an adaptation of SPR to continuous control that requires some implementation changes. The entirety of section 3.2 and the Figure 1 is exactly SPR (see Section 2.2 and Figure 2 in the SPR paper), but the paper makes no references to it except in the related work section. The related work section again fails to acknowledge that KSL and SPR share the same representation learning objective, and not only the architecture.

The authors could very well have positioned this paper as “We applied SPR to continuous control, here’s some implementation changes we needed to make along the way to make it work”, and it would have been a much more honest and accurate description of the work.  The empirical contribution here and the detailed analyses itself would have been valuable on its own.

### Questions on Experiments
------
The description of the “Generalization of Encoders” experiments is very sparse. Can you specify what exactly the training tasks and evaluation tasks are in the generalization experiment?

For the invariance experiment, it would have been nicer to see invariance to real-world distractors and not artificial noise. I would recommend the Distracting Control Suite [2] for more convincing experiments around these.

For a lot of results, the performance seems to be under-reported than the original results in papers, especially for RAD and DrQ. Here’s a link to the raw performance scores for baseline methods used in [3] https://console.cloud.google.com/storage/browser/rl-benchmark-data/dm_control, and these were reportedly obtained from the corresponding authors. Can you clarify the discrepancy in the performance data? This seems to be a major issue.

Additionally, in a lot of performance curves, the standard deviation regions overlap, making it harder to establish stochastic dominance of one method over another. It would be nicer to see a better stochastic analysis using stratified CIs on multiple normalized metrics (see Figure 11 in [3]). You can do this easily via the colab: https://bit.ly/statistical_precipice_colab

[1]  Schwarzer, M., Anand, A., Goel, R., Hjelm, R. D., Courville, A., & Bachman, P. (2020). Data-efficient reinforcement learning with self-predictive representations. ICLR 2021. https://arxiv.org/abs/2007.05929

[2] Stone, A., Ramirez, O., Konolige, K., & Jonschkowski, R. (2021). The Distracting Control Suite—A Challenging Benchmark for Reinforcement Learning from Pixels. https://arxiv.org/abs/2101.02722

[3] Agarwal, R., Schwarzer, M., Castro, P. S., Courville, A., & Bellemare, M. G. (2021). Deep reinforcement learning at the edge of the statistical precipice. NeurIPS 2021 https://arxiv.org/abs/2108.13264

**Summary Of The Paper:**

This paper shows that an auxiliary self-supervised task that enforces temporal consistency of latents improves sample efficiency in continuous control environemnts, and identifies important implementation details that make it work.

**Summary Of The Review:**

Data-efficiency of RL algorithms is an important research area, and this paper explores how auxiliary self-supervised learning can improve data-efficiency in continuous control domains. The paper is very rich with experimental details, the implementation choices are carefully ablated and the paper is overall well written and explained.

---

> ### Author Response · Authors · 2021-11-22
> **Comment for Reviewer J6YX**
>
> We thank Reviewer J6YX for their time and suggestions.
>
> - $\textbf{Similarities to SPR:}$ We agree that there are similarities between KSL and SPR, but disagree that they are exactly the same. However, as mentioned by other reviewers, the overlap harms the novelty of the contribution. As such, we will ensure that future versions of the paper further differentiate KSL from other methods by providing more detailed comparisons and ablations to highlight the impact of novel components of KSL.
>
> - $\textbf{Generalization experiment:}$ To clarify, the process for this experiment was as follows: (1) train algorithm X for 100k steps on five of the six tasks in DMControl, using five seeds for each task, and save each of the resulting 25 encoders. (2) For each of the 25 encoders, train algorithm X for 100k steps on the remaining one of the six tasks in DMControl. (3) Report evaluation results that occured during the 25 training runs in step (2). (4) Repeat for each algorithm in the list of baseline algorithms. We will make sure to include clearer language on this experiment in future versions of the paper.
>
> - $\textbf{Invariance experiment:}$ We thank Reviewer J6YX for pointing us to the Distracting Control Suite (DCS)! Considering that DCS is established within the literature, we agree that it would make for a more convincing test-bed than artificial noise for our invariance experiment. We will make sure to include DCS in future versions of the paper.
>
> - $\textbf{Reported results:}$ We produced the results in the paper using code that the authors of the chosen baselines provided. The main difference between our experiments and those in the baseline papers is the minibatch size (512 vs. 128), which we decided to keep fixed to keep comparisons fair across all algorithms.
>
> - $\textbf{Additional statistical testing:}$ Thank you for providing the links to perform the suggested additional statistical tests.

---

### Decision · Program_Chairs · 2022-01-20

**Decision:**

Reject

**Comment:**

This paper presents a reinforcement learning architecture that uses an auxiliary k-step step loss in the context of continuous control from image-based states.

While the topic is relevant and potentially impactful, several reviewers have major concerns about the manuscript. Among these, I highlight:
- Reviewers J6YX, 38iT and Qru8 have concerns about the novelty and contribution of the approach compared to existing literature.
- Reviewers J6YX, TKuY, 38iT and Qru8 have concerns about the experimental evaluation and the quality of comparisons to baselines.

Overall, it seems that the paper would benefit from further polishing.